# *Proanthocyanidins* Protect against *β-Hydroxybutyrate*-Induced Oxidative Damage in Bovine Endometrial Cells

**DOI:** 10.3390/molecules24030400

**Published:** 2019-01-22

**Authors:** Xi Cheng, Shuhua Yang, Chuang Xu, Lanzhi Li, Yi Zhang, Yang Guo, Cai Zhang, Peng Li, Miao Long, Jianbin He

**Affiliations:** 1Key Laboratory of Zoonosis of Liaoning Province, College of Animal Science and Veterinary Medicine, Shenyang Agricultural University, Shenyang 110161, China; chengxi1129@163.com (X.C.); yangshuhua0001@126.com (S.Y.); 18304076080@163.com (L.L.); sihuo12345@sohu.com (Y.Z.); jessekuo@163.com (Y.G.); 2Heilongjiang Provincial Key Laboratory of Prevention and Control of Bovine Diseases, Heilongjiang Bayi Agricultural University, Daqing 163319, China; xuchuang7175@163.com; 3College of animal science, Henan University of Science and Technology, Luoyang 471003, China; zhangcai@haust.edu.cn

**Keywords:** *proanthocyanidin*, *β-hydroxybutyrate*, oxidative stress, *Nrf2* signaling pathway, bovine endometrial cells

## Abstract

Metabolic diseases, such as ketosis, are closely associated with decreased reproductive performance (such as delayed estrus and decreased pregnancy rate) in dairy cows. The change of *β-hydroxybutyrate* (*BHBA*) concentration in dairy cattle is an important mechanism leading to ketosis, and its blood concentration in ketotic cows is always significantly higher than in nonketotic cows. Many studies indicated that *BHBA* can induce oxidative damage in liver and other organs. *Proanthocyanidins* (*PCs*) have gained substantial attention in the last decade as strong antioxidative substances. This study aimed to demonstrate a protective effect of *PCs* against *BHBA*-induced oxidative stress damage in bovine endometrial (BEND) cells by activating the *nuclear erythroid2-related factor2* (*Nrf2*) signaling pathway. Our research show that *PCs* could significantly increase activities of *catalase* (*CAT*) and *glutathione peroxidase* (*GSH-PX*), *glutathione* (*GSH*) content, and *antioxidant capacity* (*T-AOC*), while significantly decreasing *malondialdehyde* (*MDA*) content in BEND cells. Both mRNA and protein expression levels of *Nrf2* were significantly increased in BEND cells, and *glutamate–cysteine ligase catalytic subunit* (*GCLC*), heme *oxygenase 1* (*HO-1*), *manganese superoxide dismutase* (*Mn-SOD*), and *NAD(P)H*
*quinone dehydrogenase 1* (*NQO-1*) were also significantly increased. These results indicate that *PCs* can antagonize *BHBA*-induced oxidative damage by activating the *Nrf2* signaling pathway to exert an antioxidant effect.

## 1. Introduction

Ketosis occurring in dairy cows during transition period is associated with the negative energy balance (NEB). The main mechanism of NEB was the reduction of dry matter intake and the increase of energy expenditure in early postpartum period [1]. Previous studies have demonstrated that NEB triggers excessive fat mobilization and increases blood concentrations of nonessential fatty acids (NEFAs). NEFAs are first fully oxidized to provide the energy requirement for the liver and, subsequently, large quantities of NEFAs are converted to ketones (mainly *BHBA*) [2]. *BHBA* concentration in blood is always used for diagnosis of ketosis [3]. In addition, many studies have shown that *BHBA* is associated with energy metabolism, neuroprotection, and ATP production [4], and it can cause oxidative stress, inflammatory response, and even cellular apoptosis, by activating signaling pathways [5,6]. *BHBA* can efficiently activate the *Nrf2* pathway in classical *Keap1-Cys151*, and induce pathway activation in a dependent manner [7].

The structure and function of cow uterus is important for pregnancy in the next estrous cycle. Recent studies have shown that oxidative stress can affect subsequent physiological changes and metabolic functions by excessive oxidation of lipids and damage of cells in dairy cows [8]. It has been proposed that *BHBA* may play a key role in oxidative stress in dairy cows [9]. Oxidative stress is linked to uterine diseases. In humans, oxidative stress-related changes in uterine fibroid tissue samples result in infertility-related conditions, suggesting that oxidative stress plays an important role in female reproduction [10,11,12]. In rats, Marzenna showed that oxidative stress and lipid peroxidation induced by toxic Cd may affect uterus function [13]. When the balance of reactive *oxygen species* (*ROS*) and antioxidants failed, it directly affected the state of embryo and uterus in cows, resulting in loss of embryo [14]. Moreover, numerous publications reported the association of ketosis and fertility traits in dairy cows. One study found that circulating *BHBA* was increased the conception rate at first service and fetal loss was decreased in cows [15], and the calving-to-first-service interval and the calving-to-conception interval were prolonged with subclinical ketosis [16]. It was also found that a combination of NEB and reduced dry matter intake reduced fertility and milk production in dairy cows [17].

*Proanthocyanidins* (*PCs*) exhibit potent antioxidant activity due to scavenging of free radicals, and are widely present in nature [18]. *PCs* commonly exist in daily diets, and they are easy to extract, soluble in water, and suitable for in vitro assays [19]. Many studies indicated *PCs* can improve oxidative stress and degenerative diseases, including modulation of lipid and glucose metabolism [20], enhancement of innate immunity [21], and protection from neurological disorders [22], acute and chronic stress [23], and carcinogenesis [24]. The current literature confirms that, as antioxidants, *PCs* are better than vitamin E (V_E_) and have protective effects against apoptosis [25]. It was also found that feeding dairy cows grape seed and *grape marc extract* (*GSGME*) increases milk yield [26] and reduces methane emissions by affecting rumen metabolism [27], but does not influence inflammation or *endoplasmic reticulum* (*ER*) stress in the liver. In other studies, *PCs* were associated with antioxidant activity and the ability to activate the *Nrf2* defense pathway [28,29].

The purpose of our research was to reveal whether *PCs* could protect bovine endometrial (BEND) cells from oxidative stress by activating the *Nrf2* pathway. This study elucidates a protective mechanism from oxidative stress in dairy cows in transition period.

## 2. Results

### 2.1. The Effect of BHBA and PCs on the Relative Viability in BEND Cells

At all time points, relative cell viability was decreased with increasing *BHBA* concentration. However, the relative cell survival rate with 0.6 mmol/L *BHBA* was higher than in the control group. Only at 24 h of treatment did each *BHBA* concentration cause a significant difference in cell viability (*p* < 0.01) compared to the control group. Compared with 6, 12, and 24 h treatment with *BHBA*, relative cell viability increased when treated for 48 h (Figure 1).

At all time points, relative cell viability after treatment with 10 μmol/L *PCs* was higher than in the control group. Compared to the control group, the effect of 10 μmol/L *PCs* added to cells for 24 h was significant (*p* < 0.05) (Figure 2).

Compared to the control group, the relative viability of cells treated with *BHBA* at 1.2 and 2.4 mmol/L significantly decreased (*p* < 0.01), but not in the *BHBA* 0.6 mmol/L group (*p* > 0.05), whereas in the *PCs* group, viability significantly increased (*p* < 0.01). Compared with the *BHBA*-treated group, the relative cell viability in the *BHBA* 1.2 and 2.4 mmol/L + *PCs* groups were increased (*p* > 0.01, and in the *BHBA* 0.6 mmol/L + *PCs* group, viability was almost constant. After the addition of *PCs*, the cell relative viability was significantly decreased (*p* < 0.05) converted from very significantly decreased (*p* < 0.01), compared with the control group (Figure 3).

Compared with the control group, the relative cell viability in all *BHBA* groups significantly decreased (*p* < 0.01), and the relative cell viability in the *PCs* group did not increased. Compared with the *BHBA*-treated group, the relative cell viability of *BHBA + PCs* groups also showed no significant increase (*p* > 0.05) (Figure 4). Based on the above results, we chose *BHBA* concentrations of 0.6, 1.2 and 2.4 mmol/L and *PC*s concentration of 10 μmol/L for PC pretreated to BEND cells.

### 2.2. Effects of PCs and BHBA on SOD, CAT, GSH, GSH-PX, T-AOC, and MDA in BEND Cells

Compared with the control group, *CAT* activity and *GSH* content in *BHBA*-treated groups were significantly decreased (*p* < 0.05), and activities of *GSH-PX* and *SOD* in *BHBA*-treated groups were not affected significantly (*p* > 0.05), and the *T-AOC* content was decreased significantly in 2.4 mmol/L *BHBA* group (*p* < 0.05). The content of *MDA* was not significantly increased (*p* > 0.05) except in the 2.4 mmol/L *BHBA* group (*p* < 0.05). Compared with the *BHBA*-treated group, activities of *CAT* and *GSH-PX*, and contents of *GSH* and *T-AOC* in cotreated groups were all significantly increased, *SOD* activity in the cotreated group was not significantly increased except in the *PCs* group (*p* < 0.05), and the *MDA* content in the cotreated group was not significantly decreased (*p* > 0.05), except in the 2.4 mmol/L *BHBA* group (*p* < 0.05) (Figure 5).

### 2.3. Effect of BHBA and PCs on the Related mRNA Expression of Nrf2 Signaling Pathway in BEND Cells

We observed that the expression levels of *GCLC* in the 1.2 and 2.4 mmol/L *BHBA* groups was significantly increased (*p* < 0.01). By contrast, the expression levels of *HO-1* in the 2.4 mmol/L *BHBA*-treated group was significantly decreased (*p* < 0.01). However, the expression of *Mn-SOD*, *NQO-1*, and *Nrf2* were not changed significantly in all *BHBA*-treated groups. Meanwhile, expression of all genes in the *PCs* group was significantly increased (*p* < 0.01). Compared with the *BHBA*-treated group, expression levels of *GCLC*, *Mn-SOD*, *NQO-1*, and *Nrf2* in the cotreated groups were all significantly increased (*p* < 0.01 or *p* < 0.05), except *NQO-1* in the 2.4 mmol/L *BHBA* cotreated group (Figure 6). When BEND cells were treated with 2.4 mmol/L *BHBA* and *PCs* simultaneously, the expression of *HO-1* was increased significantly compared with in the 2.4 mmol/L *BHBA* group.

### 2.4. Effect of BHBA and PCs on Expression of Proteins Related to the Nrf2 Signaling Pathway in BEND Cells

The protein expression levels of *GCLC, HO-1*, and *Nrf2* in *BHBA*-treated groups were all significantly increased (*p* < 0.01), and *NQO-1* was increased significantly in the 1.2 mmol/L and 2.4 mmol/L groups, whereas protein expression of *Mn-SOD* was not changed significantly. All proteins in the *PCs*-treated group had higher expression levels. Compared with the *BHBA*-treated group, all protein expression levels in the cotreated group were significantly increased (*p* < 0.01 or *p* < 0.05), except *GCLC* in the 2.4 mmol/L *BHBA* cotreated group (Figure 7).

## 3. Discussion

Dairy cows undergo tremendous physiological challenges to the homeostatic mechanisms in the transition period leading to transition stress in the form of increased oxidative stress, reduced immunological capacity, and generation of a negative energy balance (NEB), which ultimately results in impaired postpartum fertility [30]. Ketosis in cows is usually accompanied by severe oxidative stress and inflammatory response, which were mainly associated with increased *BHBA* and *NEFA* concentration in blood [31]. Studies have shown high concentration of *BHBA* can induce oxidative stress in abomasum smooth muscle cells and calf hepatocytes, and affect the function of abomasum and liver in cows [32,33], and Amine’s research indicated high *BHBA* might pose a risk for metritis and placental retention and lead to reproductive disorders, but the association between uterus function and high *BHBA* and its mechanism was unclear.

Oxidative stress-produced *ROS* are eliminated by established antioxidant enzymes and numerous nonenzymatic defense mechanisms. In fact, the antioxidant enzymes, including *SOD*, *GSH-PX*, and *CAT*, are often measured as markers of the body’s antioxidant capacity. Additionally, numerous nonenzymatic substances protect the body from oxidative stress, e.g., *GSH* [34,35]. *MDA* acts as a marker of lipid peroxidation in the body to reflect cellular damage [36]. The levels of *T-AOC* and *MDA* directly reflect the body’s oxidation and antioxidant capacity. Our results showed that *GSH* and *CAT* were decreased significantly when cells were treated with 1.2 and 2.4 mmol/L *BHBA* whereas, in cells treated with 2.4 mmol/L *BHBA, MDA* was increased and *T-AOC* was decreased, significantly. The concentration of *BHBA* was always more than 1.2 mmol/L in subclinical ketotic cows’ blood, and might be more than 2.4 mmol/L in clinical ketotic cows’ blood. The oxidation index of the *BHBA* 2.4 mmol/L treatment group all showed significant changes in addition to *GSH-PX* and *SOD*, which indicate clinical ketotic and subclinical ketotic dairy cows were experienced varying degrees of oxidative stress, and severe oxidative stress may lead to secondary diseases in dairy cows. However, after treatment with *PCs* and *BHBA* simultaneously, the levels of the above oxidation markers changed in the opposite direction. In short, these results demonstrated that *PCs* can effectively relieve significant *BHBA-*induced oxidative stress in BEND cells, in agreement with previous studies of the antioxidant effect of *PCs* during apoptosis in kidney tissue [23]. The free radical scavenging ability of *PCs*, directed at biochemically produced *ROS* and hydroxyl radicals, has been previously reported in vitro and in vivo. For example, a treatment with *PCs* could significantly decrease *3-nitropropionic acid (3-NPA)*-induced oxidative damage in mouse ovaries [37]. Another study showed that *PCs* could act against *H_2_O_2_*-induced oxidative stress damage to protect key *fibroblast (HDF)* function by suppressing mitochondrial membrane damage in human diploid *HDFs* [38]. As previous studies have shown that cows with subclinical ketosis always have BHBA >1.2 mmol/L in their blood, compared with nonketotic cows, our results confirmed that ketotic cows always experience an oxidative stress in endometrium, which affects the function of the uterus [2].

The *Nrf2–ARE* system can exert strong antioxidative stress in cells and is an important defense mechanism of the body. *Nrf2* is a nuclear transcription factor that protects cellular homeostasis. The mechanism involves binding to *antioxidant response elements (AREs)* to activate a series of downstream antioxidant genes and proteins’ expression, such as *GCLC, HO-1, NQO-1,* and *Mn-SOD,* in response to oxidative stress [39]. Numerous publications have shown that activation of the *Nrf2* signaling pathway may help many kinds of cells to resist oxidative stress caused by *ROS*. *Orientin (Ori)*, the antioxidative matter isolated from plants, may exhibit a protective role against *H_2_O_2_*-stimulated oxidative *damage in RAW 264.7 cells by the increasing of HO-1 expression through the activation of the* Nrf2 signaling pathway [40]. Previous studies also confirmed that *nuclear factor erythroid-derived 2 (NFE2L2*), formerly known as *Nrf2*, acts in the *ARE* pathway to protect bovine mammary *epithelial cells (BMEC)* against *H_2_O_2_*-induced oxidative stress injury [41]. To further demonstrate that oxidative stress may affect BEND cells through activation of the *Nrf2* signaling pathway, we examined the mRNA and protein expression of *Nrf2, HO-1, NQO-1, GCLC,* and *Mn-SOD*. Our studies showed that when *BHBA* was applied to BEND cells alone, the mRNA expression of *Nrf*2-related genes were not changed significantly, except *GCLC* and *HO-1*, but the protein expression levels of *GCLC, NQO-1, HO-1,* and *Nrf2* (but not *Mn-SOD*) were higher, whereas for cells treated with *PCs* and *BHBA* simultaneously, all trends were enhanced compared to the *BHBA* group, consistent with previous studies [42,43]. It was important that the *GCLC* gene encoded a catalytic subunit of the *GCLC* protein and was involved in the synthesis of *GSH* [44]. The *HO-1* gene encoded the *HO-1* enzyme to protect the body from oxidative stress by regulating the elimination of toxic hemoglobin and the formation of iron ions [45]. From the experimental results, we found that the mRNA expression of most *Nrf2*-related genes in the *BHBA* treatment group were not changed significantly, such as *Mn-SOD* and *NQO-1,* and it might be that the transcriptional regulation process was also affected by many factors, such as protein translation efficiency, stability, and miRNA interference, which is a scenario worthy of further exploration [46]. What we are interested in is that the mRNA and protein expression of *Mn-SOD* was not significantly changed in all *BHBA* groups, but increased significantly in groups treated with *PCs* and *BHBA* simultaneously, and more studies are needed to confirm the function of *Mn-SOD* in oxidative stress. These findings clearly demonstrate that *PCs* can ameliorate oxidative stress in BEND cells, potentially due to antioxidative activity and the ability to activate the *Nrf2* pathway.

## 4. Materials and Methods

### 4.1. Cell Culture

Bovine endometrial cells (BEND cells) were purchased from BeNa Culture Collection (BNCC340413, Beijing, China). Cells were cultured in *DME/F-12* medium (HyClone, Logan, UT, USA), supplemented with 10% fetal bovine serum (BioInd, Beit-Haemek, Israel) and 2% *penicillin/streptomycin* (HyClone, Logan, UT, USA), at 37 °C in an incubator with 95% air and 5% CO_2_ atmosphere. The culture solution was changed each day, and subculture was conducted by trypsinization with 0.05% trypsin until the cells reached 85–90% confluence. The BEND cells were cultured in cell flasks and prepared for the following experiments.

### 4.2. PCs, BHBA Preparation and Treatment

In cows suffering from ketosis, serum *BHBA* content is used as a reference [32]. *PCs* and *BHBA* were obtained from Solarbio (Solarbio, Beijing, China). Preparation of *BHBA* was as follows: *BHBA* powder was fully dissolved in distilled water. This solution was filter-sterilized and stored at −20 °C. Preparation of *PCs* was as follows: *BHBA* powder was fully dissolved in distilled water. After filter-sterilization, this solution was stored at 4 °C. When cells reached 85–90% confluence, BEND cells were treated with serum-free media prior to incubation with *BHBA* or *PCs.*

### 4.3. Cell Proliferation Assay

*Cell proliferation assays* were performed with *Cell Counting Kit-8* (CCK-8, Solarbio, Beijing, China) according to the manufacturer’s instructions. Briefly, the cells (2 × 10^6^ cells/mL) were cultured in 96-well plates at 37 °C in a 5% CO_2_ incubator for 7–12 h, then, the cells were treated with different concentrations of *BHBA* (0–9.6 mM) and *PCs* (0–100 μM) individually for 6, 12, 24, or 48 h. After replacing with fresh serum-free medium and adding *CCK-8* reagent to 96-well plates, the cells were further incubated for an additional 1–3 h in the 37 °C incubator. The absorbance at 450 nm was measured by a microplate reader, and the cell proliferation rate was calculated according to the formula. After obtaining the optimum concentration and time, we also performed a joint treatment, including a simultaneous treatment and pretreatment, as described above.

### 4.4. Assessment of Oxidative Stress

BEND cells (2 × 10^6^ cells/mL) were seeded in 6-well plates. Confluent (90%) cells were stimulated with various concentrations of *BHBA* and *PCs* for 24 h. The supernatant was used to determine the total antioxidant capacity *(T-AOC)* and contents of superoxide dismutase (*SOD*), malondialdehyde (*MDA*), reduced glutathione (*GSH)*, glutathione peroxidase (*GSH-PX*), and catalase (*CAT*) kits (Jiancheng, Nanjing, China) according to the manufacturer’s instructions.

### 4.5. RNA Extraction and RT-PCR

The total RNA from the treated cells was isolated by using TRIzol (RNAiso Plus, TaKaRa, Dalian, China) according to the manufacturer’s instructions. RNA concentration and purity (260/280) were determined using Ultra Low-Volume Spectrometer (BioDrop, Cambridge, UK). Approximately 5 µg of each sample of total RNA was reverse-transcribed to cDNA with PrimeScript RT reagent kit (TaKaRa, Tokyo, Japan), as described in the manufacturer’s protocol. cDNA was stored at −20 °C until it was used for real-time PCR. The primers for RT-PCR were designed by Primer 5 software (IBM, Almon, NY, USA), and are listed in Table 1.

mRNA expression levels were measured by using SYBR Premix Ex Taq II (TaKaRa, Dalian, China) on the QuantStudio 3 gene amplification instrument (ABI, Waltham, MA, USA) real-time PCR analysis.

The cycling conditions were as follows: 95 °C for 30 s, and then 40 cycles of 95 °C for 5 s, 60 °C for 34 s, and 60 °C for 1 min. All amplifications were repeated three times. To analyze the relative level of expression of each mRNA, the melting curves were used to analyze and assess the accuracy of the PCR and the 2^−ΔΔCt^ values were used to quantify gene expression.

### 4.6. Western Blotting

Total proteins were obtained from the treated BEND cells with a protein extraction kit (Solarbio, Beijing, China). The protein concentration was detected by the BCA method (Solarbio, Beijing, China). The proteins (50 μg) were heated in loading buffer at 95 °C for 5 min. Each protein sample was subjected to SDS-PAGE using a gel preparation kit (Solarbio, Beijing, China) to separate proteins, then proteins were transferred to the PAGE membrane and blocked with 5% skim milk at 4 °C overnight. After rinsing with *TBST* on a shaker, membranes were hybridized for 90 min at room temperature with anti-*Nrf2* (1:2000, Abcam, Cambridge, UK), anti-GCLC (3:1000, Abcam), anti-HO-1 (3:1000, Abcam), anti-NQO-1 (1:2000, Novus, Centennial, CO, USA), anti-*Mn-SOD* (1:5000, Abcam), and anti-*β-actin* (1:1000, Cell Signaling) antibodies, respectively. After rinsing for 1 h, membranes were incubated with corresponding horseradish peroxidase-conjugated secondary antibodies for 90 min at room temperature. After rinsing again for 1 h, proteins were visualized using *Enhanced Chemiluminescence (ECL)* system. Finally, an image-analysis system was used to analyze the density of these target proteins.

### 4.7. Statistical Analysis

All values are expressed as the means ± SEM. The statistically significant differences were assessed by analysis of variance (*ANOVA*) followed by Tukey’s multiple comparisons test using GraphPad Prism v. 5.0 for Mac (GraphPad Software, La Jolla, CA, USA). Statistical significance was defined as *p* < 0.05 or *p* < 0.01.

## 5. Conclusions

In summary, our study confirmed that *BHBA* could cause oxidative stress in BEND cells and might be the key factor leading to reproductive disorders in ketonic cows, whereas *PCs* can relieve the oxidative damage by activating the *Nrf2* signaling pathway.

## Figures and Tables

**Figure 1 molecules-24-00400-f001:**
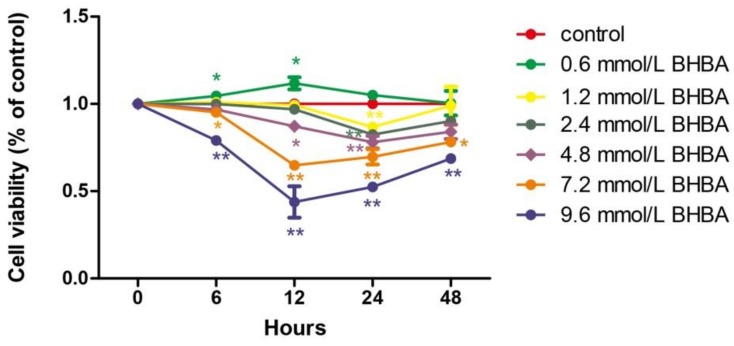
The effect of different of times and *β-hydroxybutyrate* (*BHBA*) concentrations on the relative viability of bovine endometrial (BEND) cells. * *p* < 0.05 vs. control group, ** *p* < 0.01 vs. control group.

**Figure 2 molecules-24-00400-f002:**
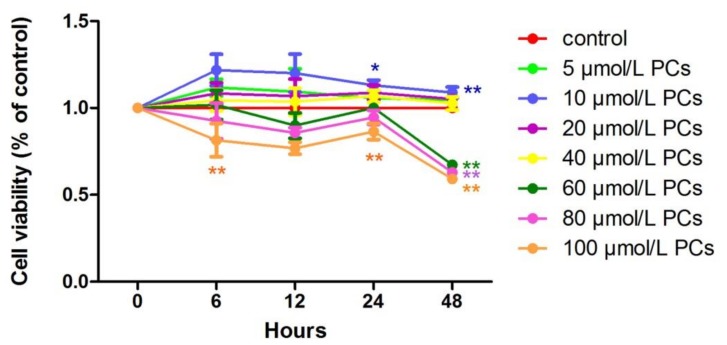
The effect of different of times and concentrations of *proanthocyanidins* (*PCs*) on the relative viability of BEND cells. * *p* < 0.05 vs. control group, ** *p* < 0.01 vs. control group.

**Figure 3 molecules-24-00400-f003:**
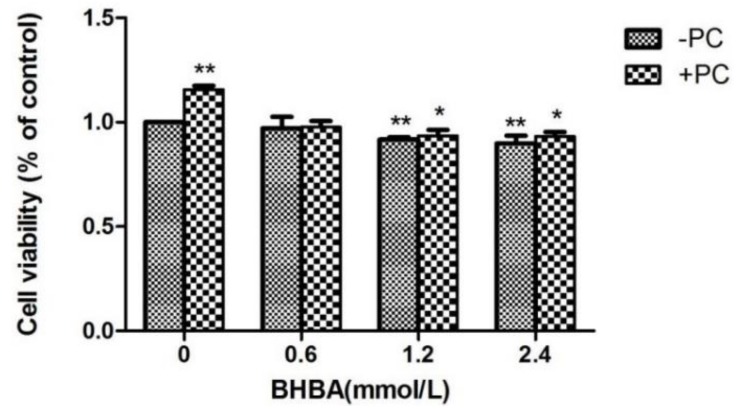
Effects of different *BHBA* concentrations and fixed *PCs* concentration of 10 µmol/L on relative viability of simultaneously treated BEND cells. * *p* < 0.05 vs. control group, ** *p* < 0.01 vs. control group.

**Figure 4 molecules-24-00400-f004:**
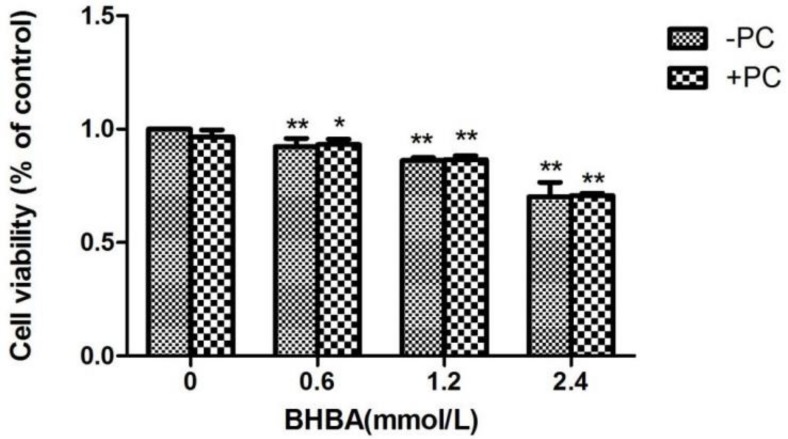
Effects of different *BHBA* concentrations and *PCs* pretreatment on the relative viability of BEND cells. * *p* < 0.05 vs. control group, ** *p* < 0.01 vs. control group.

**Figure 5 molecules-24-00400-f005:**
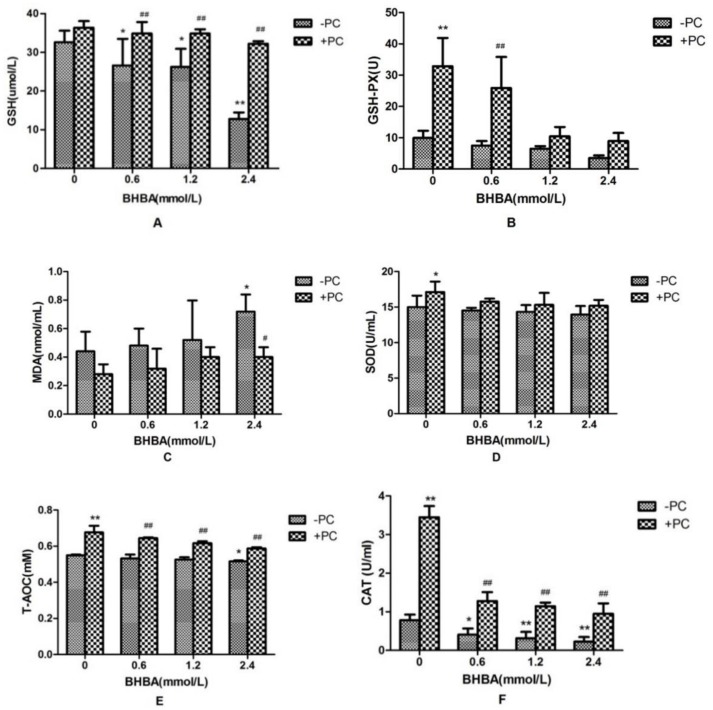
Effects of *BHBA* and *PCs* on the antioxidation markers in BEND cells. (**A**) The *GSH* content. (**B**) The *GSH-PX* activity. (**C**) The *MDA* content. (**D**) The *SOD* activity. (**E**) The *T-AOC.* (**F**) The *CAT* activity. * *p* < 0.05 vs. control group, ** *p* < 0.01 vs. control group. # *p* < 0.05 vs. BHBA-treated group, ## *p* < 0.01 vs. BHBA-treated group.

**Figure 6 molecules-24-00400-f006:**
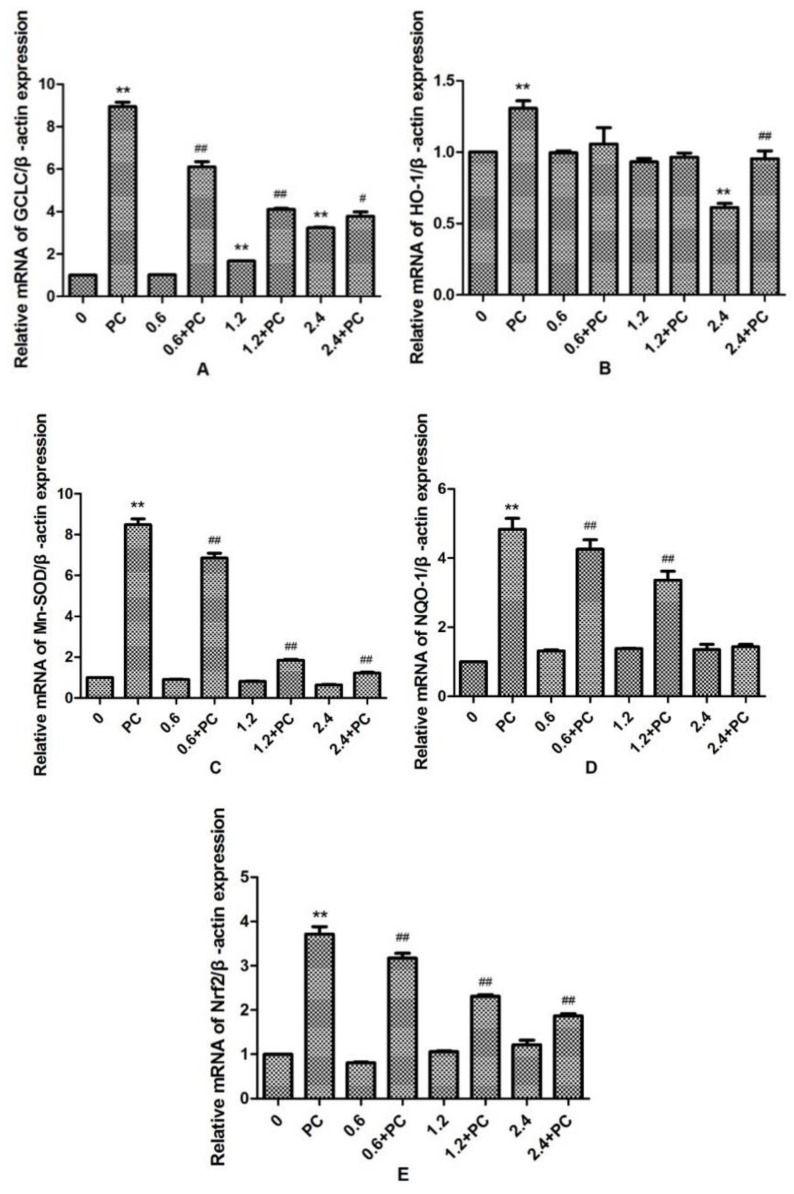
The effect of simultaneous treatment with *BHBA* and *PCs* on the mRNA expression of genes related to the *Nrf2* signaling pathway in BEND cells. ** *p* < 0.01 vs. control group. # *p* < 0.05 vs. *BHBA*-treated group, ## *p* < 0.01 vs. *BHBA*-treated group.

**Figure 7 molecules-24-00400-f007:**
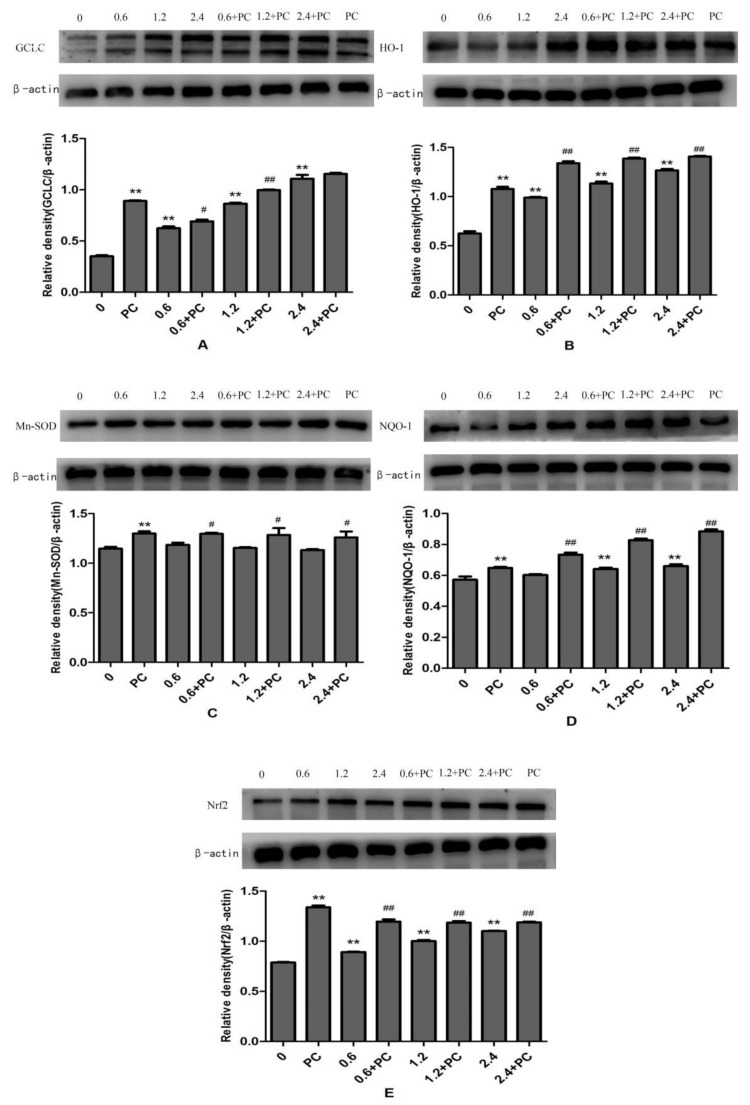
The effect of simultaneous treatment with *BHBA* and *PCs* on the protein expression of *GCLC, HO-1, Mn-SOD, NQO-1*, and *Nrf2* in BEND cells. ** *p* < 0.01 vs. control group. # *p* < 0.05 vs. *BHBA*-treated group, ## *p* < 0.01 vs. *BHBA*-treated group.

**Table 1 molecules-24-00400-t001:** Parameters for chemical structures.

Gene	GenBank Accession NO.	Primer Sequence (5′–3′)	Fragment Size
*Nrf2*	AC_000159.1	Forward: AGCGGCTTGAATGTTTGTCTT	130 bp
		Reverse: CCCAGTCCAACCTTTGTCGTC	
*Mn-SOD*	AC_000166.1	Forward: AGTTGACTGCTGTATCTGTTGGTGTC	239 bp
		Reverse: GGTATGAACAAGCAGCAATCTGTAA	
*HO-1*	NM_001014912.1	Forward: AATATCGCCAGTGCCACCAAGTTC	142 bp
		Reverse: GTTGAGCAGGAAGGCGGTCTTG	
*GCLC*	NM_001083674.1	Forward: CACCACGAACACCACATACGC	198 bp
		Reverse: ACCTGGATGATGCCAACGAGT	
*NQO-1*	AC_000175.1	Forward: GCTACTTGGAGCAAAATACAG	204 bp
		Reverse: CTTGGAACCTCAACTGACATA	
*β-actin*	AC_000182.1	Forward: GCCCTGAGGCTCTCTTCCA	101 bp
		Reverse: GCGGATGTCGACGTCACA	

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
