# Peer review of "Proanthocyanidins Protect against β-Hydroxybutyrate-Induced Oxidative Damage in Bovine Endometrial Cells"

_molecules, 2019, doi:10.3390/molecules24030400_

Round 1

Reviewer 1 Report

The authors are trying to establish the protection role of proanthocyanidins in hydroxybutyrate induced oxidative damage in BEND cells.

1) In Line 117 the authors claimed to be decrease in SOD and T-AOC levels but is not true in fig 5 (D and E).

2) In Line 133 the authors claimed to be decrease in Mn-SOD level but is not true in fig 6 C.

3) In Line 203 the increase and decrease as authors claimed are not reflecting in Figs [eg: Fig 7 (C and D).
4) The size of Fig 5 should be increase.

Author Response

Dear reviewer:

I am very grateful to your comments for the manuscript. According with your advice, we amended the relevant part in manuscript. Some of your questions were answered below.

1)  The reviewers comment: In Line 117 the authors claimed to be decrease in SOD and T-AOC levels but is not true in fig 5 (D and E).

The author’s answer: I have already revised it in the manuscript, In Line 119, I have re-described the results accurately. From the data analysis, the levels of SOD and T-AOC were decrease, but not significant (p>0.05).

2)  The reviewers comment: In Line 133 the authors claimed to be decrease in Mn-SOD level but is not true in fig 6 C

The author’s answer: I have already revised it in the manuscript, In Line 138, I have re-described the results accurately. the expression of Mn-SOD, NQO-1, Nrf2 were not changed significantly in all BHBA treated groups (p>0.05).

3)  The reviewers comment:  In Line 203 the increase and decrease as authors claimed are not reflecting in Figs [eg: Fig 7 (C and D).
    The authors answer: I have already revised it in the manuscript, In Line 214, I have re-described the results accurately. Our studies showed that when BHBA was applied to BEND cells alone, the mRNA expression of Nrf2-related genes were not changed significantly except GCLC and HO-1, but the protein expression levels of GCLC, NQO-1, HO-1, Nrf2 were higher except Mn-SOD, whereas treated cells with PC and BHBA simultaneously, all trends were enhanced compared to BHBA group.

3)  The reviewers comment: The size of Fig 5 should be increase.

The author’s answer: I have increased the size of Fig 5 in Line 127.

Reviewer 2 Report

This manuscript investigated the molecular mechanisms through which proanthocyanidins can modulate the BHBA-induced oxidative damage evaluating the Nrf2 signaling pathway and the antioxidants levels and MDA as marker of oxidative stress. The results indicate that PCs activating the Nrf2 pathway to exert the antioxidant effect.

The paper is well written but there are some aspects in results need to be checked:

- In line 101 the authors said that viability significatly decreased (Figure 3) but the decreased not is significative por 0.6 mmol/L.

- In line 130 the authors indicate that the expresion levels of GCLC, NQO-1 and Nrf2 in BHBA-treated groups increased with increasing BHBA concentrations. However, in figure 6 only there are significance in the expression of GCLC.

In material methods the authors could indicate the yield of PCs after to filter and the concentration used to treated the cells. The authors have used a kit to evaluate the MDA levels, but Are sure that measured MDA and not TBARs?

The discussion is poor, the results no were discussed reasonably.  The discussion is a description of results. It is nec

Author Response

Dear reviewer:

I am very grateful to your comments for the manuscript. According with your advice, we amended the relevant part in manuscript. Some of your questions were answered below.

1)  The reviewer’s comment: In line 101 the authors said that viability significatly decreased (Figure 3) but the decreased not is significative por 0.6 mmol/L.

The author’s answer: I have already revised it in the manuscript, In Line 98, I have re-described the results accurately. Compared to the control group, the relative viability of cells treated with BHBA at 1.2 and 2.4 mmol/L significantly decreased(p<0.01), but not in the BHBA 0.6mmol/L group (p>0.05). 

2)  The reviewers comment: In line 130 the authors indicate that the expresion levels of GCLC, NQO-1 and Nrf2 in BHBA-treated groups increased with increasing BHBA concentrations. However, in figure 6 only there are significance in the expression of GCLC.

   The author’s answer: I have already revised it in the manuscript, In Line 137, I have re-described the results accurately. We observed that the expression levels of GCLC in the 1.2 and 2.4 mmol/L BHBA groups was significantly increased (p<0.01), the expression levels of HO-1 in the 2.4mmol/L BHBA-treated group was significantly decreased (p<0.01). But the expression of Mn-SOD, NQO-1, Nrf2 were not changed significantly in all BHBA treated groups. 

3)  The reviewers comment: In material methods the authors could indicate the yield of PCs after to filter and the concentration used to treated the cells. The authors have used a kit to evaluate the MDA levels, but Are sure that measured MDA and not TBARs?

   The author’s answer: PCs was obtained from Solarbio (Solarbio, Beijing, China).PCs is completely dissolved at 10mg/mL water, which can completely through the 0.22μm filter membrane, after to filter not affect the concentration of PCs, which is consistent with the previous results[1]. When cells reached 85-90% confluence, BEND cells were treated with PC diluted with DME / F-12.

The kit for detecting MDA used by me was purchased from Jiancheng(Nanjing, China)and the item number is A003-1. It is currently the most stable MDA test method in China, which is consistent with the previous results[2, 3]. 

1. Long, M.; Chen, X.; Wang, N.; Wang, M.; Pan, J.; Tong, J.; Li, P.; Yang, S.; He, J., Proanthocyanidins Protect Epithelial Cells from Zearalenone-Induced Apoptosis via Inhibition of Endoplasmic Reticulum Stress-Induced Apoptosis Pathways in Mouse Small Intestines. 2018.

2. Wang, N.; Li, P.; Wang, M.; Chen, S.; Huang, S.; Long, M.; Yang, S.; He, J., The Protective Role of Bacillus velezensis A2 on the Biochemical and Hepatic Toxicity of Zearalenone in Mice. Toxins.

3. Yang, S. H.; Yu, L. H.; Li, L.; Guo, Y.; Zhang, Y.; Long, M.; Li, P.; He, J. B., Protective Mechanism of Sulforaphane on Cadmium-Induced Sertoli Cell Injury in Mice Testis via Nrf2/ARE Signaling Pathway. Molecules 2018, 23, (7), 1774-.

Round 2

Reviewer 1 Report

line 118: According to figure caption Fig 4 is PC pretreated cells. Change "simultaneous addition" in line 118 to "PC pretreated".

lines 100 to 125:  check word spaces.

line 155, 168, 268: check word spaces

Reviewer 2 Report

The author has answered the questions correctly. The manuscript has improved with respect to the first version.